:ֺ҉: PLOS | ONE

# The role of scientific communication in predicting science identity and research career intention

**Carrie Cameron**[1]*, **Hwa Young Lee**[1], **Cheryl B. Anderson**[1], **Jordan Trachtenberg**[2], **Shine Chang**[1]

**1** Division of Cancer Prevention and Population Sciences, Cancer Prevention Research Training Program, The University of Texas MD Anderson Cancer Center, Houston, Texas, United States of America, **2** Office of Institutional Research, Planning, and Assessment, Rose-Hulman Institute of Technology, Terre Haute, Indiana, United States of America

* ccameron@mdanderson.org

**Data Availability Statement:** All relevant data are within the manuscript and its Supporting Information files.

## Abstract

The number of biomedical sciences PhDs persisting in academic faculty careers has been declining. As one potential influence on trainees' intention to persist, we investigate the development of scientific communication (SC) skills, hypothesizing that attitudes and behaviors regarding scientific writing, speaking, and presenting predict academic research career intention, through science identity. After adapting a social-cognitive career theoretical model of SC to include science identity and mentor practices, we conducted a longitudinal survey of 185 doctoral and postdoctoral fellows. Structural equation modeling was used to examine relationships among SC productivity, SC self-efficacy, SC outcome expectations, mentor practices in SC, science identity, and research career intention. Results confirmed the overall model and revealed additional specific pathways: SC productivity and SC outcome expectations directly predicted career intention; SC productivity and mentor practices predicted science identity through SC self-efficacy. Demographic factors did not predict intention when controlling for SC variables. Findings support a model of SC skill development as a predictor of research career intention ($R^2 = .32$). The finding that SC language use predicts science identity has important sociolinguistic implications. The key factors in this process are actionable at the trainee, mentor, and institutional levels, suggesting potential for SC interventions to increase career persistence.

## Introduction

The number of PhDs in biomedical sciences entering academic careers with the goal of becoming independent investigators has been declining in recent years. Roach and Sauermann [1] found that 25% of STEM doctoral students lost interest in an academic research career over the course of their PhD. A variety of causes has been identified to explain this trend; some of the structural causes include diminished funding opportunities, insufficient faculty positions to absorb the numbers of new PhDs, and the emergence of new types of research positions and

**Funding:** The study referred to in this article was supported by National Institute of General Medical Sciences (https://www.nigms.nih.gov/) grant R01GM085600 (C.C. & S.C., MPIs, Retention in Research Careers: Mentoring in Scientific Communication). The content is solely the responsibility of the authors and does not necessarily reflect the views of the funding agency. The funding agency had no role in study design, data collection and analysis, decision to publish, or preparation of the manuscript.

**Competing interests:** The authors have declared that no competing interests exist.

opportunities outside academia [2–4]. At the same time, the low retention of women, especially women of historically under-represented minority (URM) groups, in academic research tracks, decreases consistently and disproportionately toward the later stages of training and throughout faculty career progression and is a cause for concern [5, 6], limiting efforts to build a diversified biomedical workforce.

An ecological approach to understanding the declines in pursuit of faculty research careers, as advocated by Gibbs [6], takes into account multiple factors, such as social and psychological factors, from the individual to the policy level. On the individual level, research on these mechanisms has focused on several key psychological and social mediators. Among undergraduates, post-baccalaureate students, or early-stage graduate students pursuing STEM tracks, these include science identity, sense of belonging in the community of science or cultural congruence with the institutional environment [7, 8], research self-efficacy (perception that one is competent at research, e.g., [9, 10]), and science values (sharing the values of the scientific community; [11]). While the PhD and postdoctoral levels have received less attention, Gibbs found that factors influencing interest in faculty careers among postdocs include research self-efficacy, mentor's investment in their career advancement, high interest in a faculty career at the time of PhD completion, and first-author publication rate [12].

Ironically, while first-author publication rate is a time-honored yardstick for measuring success in academia, the precursors to success in becoming a published author are rarely studied. The present study extends the literature on drivers of research career intention by examining scientific communication skill development and its longitudinal effects. Because language use and identity are interrelated, as we discuss below, we sought to understand how SC skills and science identity may predict career intention.

## Language, identity, and scientific communication

The current study focused on the development of scientific communication (SC) skills, defined as writing, formal oral presentation, and spontaneous speaking (conversing, asking questions, etc.), and science identity. An earlier cross-sectional study found that at the PhD and postdoctoral level engagement in SC and social cognitive variables were predictive of intention to pursue an academic research career [13]. The longitudinal study reported here builds on this work by testing temporal relations between these variables, with the addition of science identity.

All aspects of language—accent, intonation, grammar, vocabulary, conversational style—are markers of identity [14–18]. We identify almost instantaneously whether someone is a member of our in-group by the way they speak or write; this has been observed even in preverbal children [19]. Likewise, we signal our identity to others by the learned habits of speech and writing we exhibit and by the choices we make of what kinds of language to use in which contexts use.

Language and its varieties are learned and developed in a social context through prolonged interaction with others in the environment. (A *language variety* is a different version of the same language, such as a regional dialect.) A tenet of socio-cultural linguistic scholarship holds that identity emerges from accumulated linguistic interactions, rather than existing as a prior psychological entity independent of linguistic interaction [20, 21]. Scientific communication, although often thought of popularly as merely the use of some specialized vocabulary, is understood in linguistic terms as a distinct language variety, characterized by identifiable grammatical, lexical, and stylistic features (see [22] for a more detailed treatment of the features of scientific communication).

Like other language varieties, SC is acquired gradually, through extended exposure to a community of professional practice [23–25]. Mastery is achieved through sustained

interaction, both written and spoken, with surrounding senior and peer researchers. As Flowerdew and Wang note,

> In the field of academic research. . .textual production [i.e., writing] is at the core of negotiating the interactive relationships among the members of academic communities and claiming and constructing identities. . ..from submitting manuscripts to journals to writing grant proposals, junior researchers have to satisfy the expectations of their more senior colleagues in order to climb up the ladder. . .. Such activities involve, among other things, a process of identity transformation and academic acculturation [26, p.82].

Additional scholarship has focused on the intersection of identities of race, ethnic culture, and gender with academic identity, as trainees are socialized into their respective disciplines [6, 27–30]. This interdependent relationship between language use in the research environment and identity underlies the present research. Grounded in the socio-cultural conception that language use is a creator of identity as opposed to being a product of identity, we considered SC as a possible source of science identity and thus ultimately as a predictor of career intention and persistence.

## Modes of scientific communication

*Scientific communication* in the scholarly sense (as opposed to *science communication*, which concerns communication with the public) is usually thought of as finished research products such as manuscripts and conference presentations [13]. By contrast, the cornerstone of the linguistic approach taken here is the conception of SC as a broad spectrum of communication activities including formal scientific writing and presenting as well as informal *speaking* or incorporating everyday conversations in the research environment. Speaking differs from writing and presenting in that conversations cannot be planned, rehearsed, or edited and involve the risk of exposing vulnerabilities. Activities such as making comments or asking questions in group discussions, conference presentations, or lectures, and participating in journal clubs or other gatherings are common examples of informal speaking in academic environments. The role of speaking as an SC construct, distinct from writing and presenting, has been psychometrically validated [13, 31].

## Theoretical framework

We used social cognitive career theory (SCCT; [32]) as a model framework to investigate scientific communication skills as predictors of intention to pursue research careers. Briefly, SCCT posits that an individual's *self-efficacy* or sense of competence in a particular domain, and *outcome expectations*, the anticipated consequences of these behaviors, foster interest and goals for pursuing a career in a particular domain. SCCT also describes how *supports* and *barriers*, as contextual factors, affect this process by facilitating or inhibiting the vocational outcomes [33]. Four sources of learning are posited to influence one's level of self-efficacy: *performance accomplishments* (e.g., mastery experiences with specific tasks); *vicarious learning* (e.g., observing the behavior of others, modeling, such as showing one how to perform a research procedure or task); *social persuasion* (e.g., verbal encouragement); and *affective and physiological states* (e.g., positive or negative states, such as anxiety, related to performing a task; [34]).

Our previous cross-sectional study with biomedical sciences doctoral and postdoctoral trainees extended the SCCT model by examining the role of scientific communication as an ancillary, but necessary skill in the routine activities of a successful research career. Although writing, presenting, and speaking are unlikely to be what attracted the individual to the career

to begin with, they are indispensable activities for career success. This is in contrast to core career-specific skills, such as lab and study-design skills. Central constructs (see Table 1 for definitions) in this study were validated, and trainees' SC outcome expectations and SC productivity (i.e., frequency of writing and speaking) were found to directly predict intention to pursue a research career. Self-efficacy in scientific communication was found to indirectly predict such intentions, through SC outcome expectations.

## The current longitudinal study

To further substantiate and extend our previous model, we designed a longitudinal and dyadic survey-based study of mentors and their doctoral and postdoctoral mentees, allowing us to observe the behavior of the variables over four time points and within specific mentoring relationships. Two new variables of interest were introduced into the trainee surveys during this phase. The first was *science identity*, or the sense of being a "science person." Carlone and Johnson characterize science identity as including the dimensions of competence, performance (e.g., "ways of talking"), and recognition by others as a scientist and illustrate with the example of a researcher "presenting her work at a conference [who] must use language according to prescribed norms" [35, p.1190].

Science identity has been proposed and supported as a mediator between self-efficacy and outcome expectations in recent scholarship on STEM career outcomes of undergraduate students. Using the tripartite integration model of social influence, Estrada found that science self-efficacy was associated with science identity, but did not predict career outcomes [11]. A recent SCCT path model of STEM major choice among undergraduate students found that science identity indirectly predicted career intention through outcome expectations [36]. Similarly, we included science identity as a potential mediator between SC self-efficacy and career intention in our longitudinal model.

The second variable introduced was *mentor practices in SC*, as reported by trainee respondents, to operationalize two of the four sources of learning (vicarious learning and social persuasion), in the model. Mentor practices in the development of research skills have been shown to be an important predictor of research self-efficacy [8, 11, 36, 37]. Recent SCCT studies have tested hypotheses of causal and reciprocal relationships among SCCT variables [38]. Considering the directional relationships among SCCT variables found in previous cross-sectional and longitudinal studies supporting a predictive hypothesis to explain the process through

**Table 1. Definitions of central constructs.**

| Key Term | Definition |
|---|---|
| SC productivity | Scientific communication productivity is the *frequency* with which a trainee had engaged in research-related writing, presenting (planned, rehearsed talks about research to an audience, whether at a major scientific meeting or a journal club), and speaking (asking questions after presentations, describing posters and answering questions) at Time 1. This operationalizes p*erformance accomplishments* in the SCCT model. (Note: SC Productivity does not refer to number of accepted articles or abstracts.) |
| SC self-efficacy | Individual's level of confidence in ability to compose abstracts, manuscripts, etc; to give research talks to varying audiences in various settings; to contribute questions or comments in research settings or to network in professional settings using the expected scientific style. |
| SC outcome expectations | The consequences the trainee expects to result from scientific writing and speaking. |
| Science identity | The sense of belonging to the research community; thinking of oneself as a scientist. |
| SC mentoring practices | The frequency with which the trainee reports that their mentor engages in providing encouragement and assistance with writing, presenting, and participating in scientific discussions. SC mentoring operationalizes the constructs of *vicarious learning* and *social persuasion* in SCCT models. |

which students develop career intentions, our model hypothesized that mentoring practices (as an operationalization of vicarious learning and social persuasion) and trainee SC productivity (as an operationalization of performance accomplishments) would have direct, positive effects on SC self-efficacy at Time 1. Incorporating findings regarding the inclusion of science identity and its temporal sequence within the SCCT framework [9, 36], science identity at Time 2 was hypothesized to mediate the relationship between SC self-efficacy at Time 1 and SC outcome expectations at Time 3, impacting trainee research career intention at Time 4 (see Fig 1).

The research questions for the study were: Are SC self-efficacy, SC productivity, and SC mentoring practices sources of science identity? Does science identity predict career intention? Specifically, we hypothesized that

1. the previously tested social-cognitive variables of SC self-efficacy and outcome expectations, as well as SC productivity, are significant, direct predictors of career intention over time;

2. SC self-efficacy is a direct source of science identity, and SC productivity and SC mentoring practices are indirectly related to identity, through self-efficacy;

3. science identity directly predicts SC outcome expectations;

4. science identity predicts intention to pursue an academic research career, directly and indirectly, through outcome expectations.

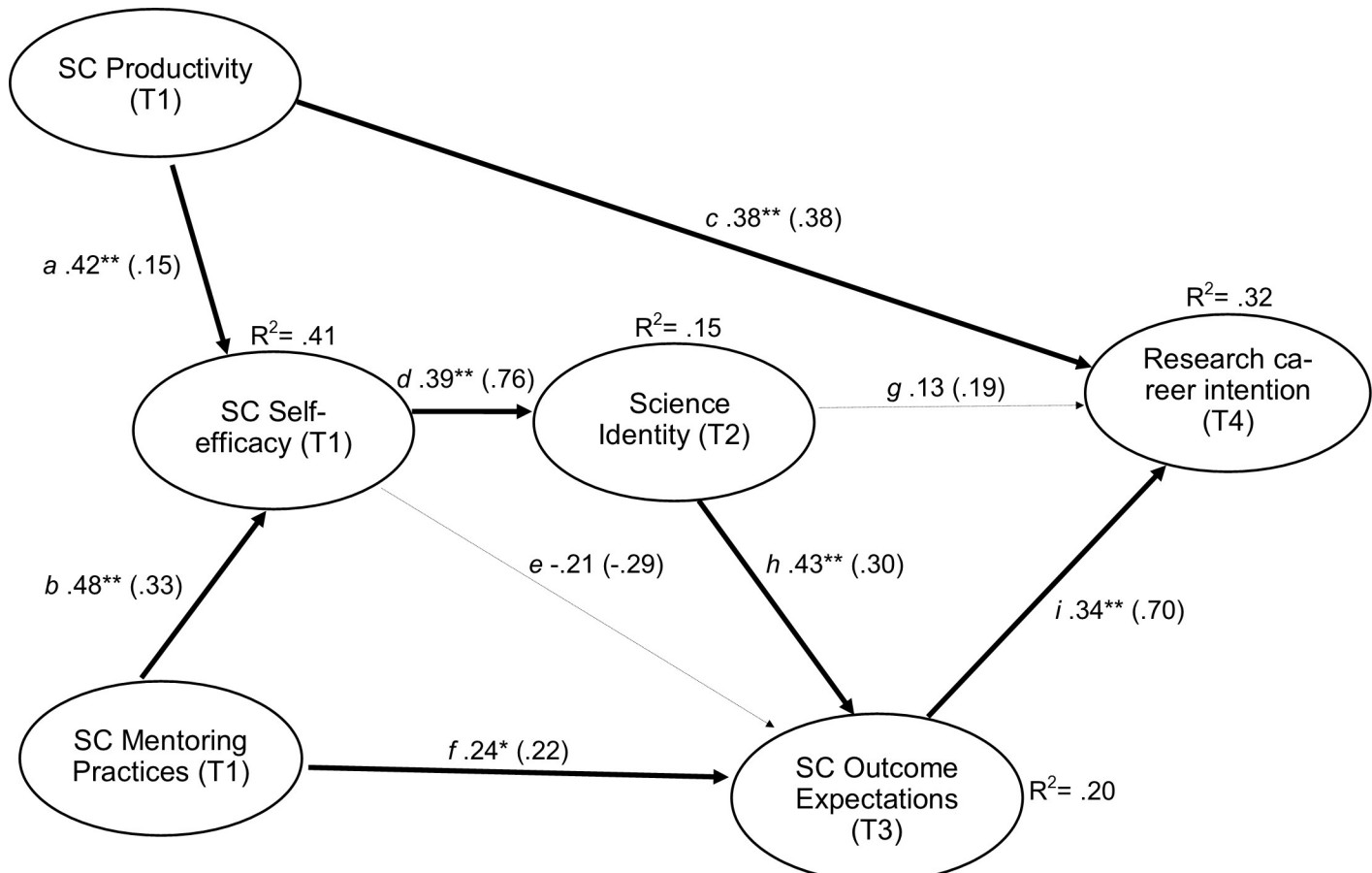

**Fig 1. Structural model.** The SC model fit the data well, and most hypothesized paths were statistically significant. $^{*}$ and $^{**}$ represent $p < .05$ and $p < .01$, respectively. Values represent standardized (unstandardized) coefficients.

## Methods

The study was conducted at four time periods with dyads of faculty mentor and trainee participants. The results in the current report concern trainee data only.

### Participants

Participants were doctoral and postdoctoral trainees in biomedical and behavioral sciences at 71 institutions throughout 33 states in the USA. To recruit trainees to the study, faculty at institutions with diverse student-body populations such as Hispanic-Serving Institutions and Historically Black Colleges and Universities were contacted first and invited to participate by email. Contacts at institutions associated with our research, mailing lists of relevant research networks, and flyers distributed at professional meetings were also used to recruit mentors, who were considered eligible for the study if they were currently mentoring one or more doctoral or postdoctoral trainees. Potential mentor participants were also asked to forward information about our study to acquaintances who might be interested in participating. Recruiting the mentors of the dyad first allowed us to avoid a scenario in which a trainee wished to participate but their mentor declined to participate as part of the required dyad. Participating mentors nominated a list of trainees as candidates for participation in the study (or a single trainee if multiple trainees were not available), and the study team then contacted a trainee from the list to email for recruitment. If the trainee declined, the study team contacted another nominee to invite. Nominating mentors were blind to who was selected from the list and whether the invited trainee agreed to participate, as well as to trainees' response data. Mentors or trainees whose primary focus was research were eligible to participate; full-time clinical mentors or trainees were excluded. The order in which eligible trainees were selected from the nominating mentors' lists was determined by demographic categories, in order to balance subgroups to the extent possible. The study was conducted from August 2014 to November 2016 and was approved by the IRB of MD Anderson Cancer Center (Protocol 2013–0829). Trainee participants that met eligibility completed online informed consent and were sent four online surveys (Time 1 –Time 4) over the two-year study.

Of the 208 mentors who expressed initial interest to participate in our study, 185 mentors had matched trainees who responded to our email contact. Thus, the sample included 185 trainee participants at Time 1. Three trainees completed only the demographic section, so were excluded from the analysis. Of the 182 remaining trainees, 93% were retained to complete the survey at Time 2, and 84% at Time 4 (16% attrition rate over time). About 6% of participants reported that they were no longer in their paired mentoring relationship at Time 4. We performed a sensitivity analysis to compare results with and without participants who reported that their mentoring relationship had changed. Preliminary analysis demonstrated that the results did not differ, so all participants were included in the analyses. Demographics of trainee participants are shown in Table 2. The mean age of the sample was 28 years (SD = 4.5; range = 22–46 years), and most participants were doctoral trainees (71%). Women comprised 60% of the sample, and participants self-identified as members of the following ethnic groups: African American (12%), Asian or Asian American (23%), White (50%), and Hispanic (13%). The majority of participants were native speakers of English (76%), and about 31% of participants were first-generation college students. Approximately half of the participants were pursuing degrees related to basic science (52%), 14% in population science, 9% in clinical science, and 25% pursued other STEM, social, or behavioral science disciplines, as indicated in an "other" category. Approximately half of the participants were not actively seeking employment (55%) at Time 4.

**Table 2. Participant characteristics (N = 185).**

| | Category | Frequency | Percentage |
|---|---|---|---|
| Gender | Female | 112 | 60.5 |
| | Male | 73 | 39.5 |
| Citizenship | U.S. Citizen | 141 | 76.2 |
| | Visa Holder | 44 | 23.8 |
| Ethnicity | Hispanic | 24 | 13 |
| | Non-Hispanic | 159 | 85.9 |
| | Don't Know | 2 | 1.1 |
| Race | Asian or Asian-American | 42 | 22.7 |
| | Black or African American | 23 | 12.4 |
| | White | 92 | 49.7 |
| | American Indian or Alaska Native | 2 | 1.1 |
| | Other | 26 | 14 |
| Primary Language | English | 139 | 76 |
| | Other Languages | 44 | 24 |
| Academic Rank | Postdoctoral Fellow | 54 | 29.2 |
| | Doctoral Student | 131 | 70.8 |
| Discipline* | Basic Science | 95 | 51.9 |
| | Population Science | 25 | 13.7 |
| | Clinical Science | 17 | 9.3 |
| | Social Science | 20 | 10.9 |
| | Engineering Science | 7 | 3.8 |
| | Other | 19 | 10.4 |
| Years in the U.S. | Less than 3 years | 19 | 10.4 |
| | 3–5 years | 15 | 8.2 |
| | 6–10 years | 19 | 10.4 |
| | 11–15 years | 4 | 2.2 |
| | Over 15 years | 126 | 68.9 |
| 1st Generation College** | Yes | 56 | 30.6 |
| | No | 125 | 68.3 |
| | Don't Know | 2 | 1.1 |
| Economic Standing*** | Far Below Average | 9 | 4.9 |
| | Below Average | 30 | 16.4 |
| | Average | 74 | 40.4 |
| | Above Average | 59 | 32.2 |
| | Far Above Average | 11 | 6 |

The total number may not reflect N = 185 due to missing data.

*Discipline was broken down into categories defined by ABRCMS. The "other" category includes disciplines where there were 2 or fewer people.

**From the survey: Are you or any of your brothers and sisters in the first generation of your family to attend college?

***Participants were asked to respond subjectively to this item, since many may be unable to specify dollar amounts.

## Procedure

The four surveys were administered online to trainees at approximately equal intervals, based on trainees' schedules (6- to 7-month window). Each survey took 20–30 minutes to complete, and trainees received a $40 gift card after completion of each survey and an additional $10 and $30 gift card if their mentors completed the corresponding mentor survey at the 1st and 4th time points, respectively.

The online surveys included demographic questions along with measures of social cognitive and behavioral variables. Past trainee SC productivity and perceived mentoring practices toward trainees in SC were measured at Time 1, trainee SC self-efficacy and SC outcome expectations were measured at Time 1–4, science identity was measured at Time 2–4, and academic career intention was measured at Time 4.

## Measures

**SC productivity.** At Time 1, trainees were asked to write in the number of SC tasks they had completed in their career to date [13, 39]. These included two types of writing tasks (e.g., "Prepared by myself a full first draft of a first-author manuscript [does not have to have been submitted]") and two presentation-task items (e.g., "Given an oral presentation at a national scientific meeting" and "How many times have you presented a poster at a scientific meeting?"). A confirmatory factor analysis was performed to ensure the validity and reliability of the six items. Two items related to speaking and presenting did not perform well on the latent construct of SC productivity, so four items were used for the analysis (two writing and two presenting). The final model with one residual covariance had an acceptable fit to our current data, $\chi^2_{S-B}$ (1) = 1.102, $p >$ .05, CFI = .999, TLI = .996, RMSEA = .024, 90% CI (.000, .203), and SRMR = .009. Cronbach's alpha was .78. SC productivity was measured only at Time 1, consistent with SCCT models which measure performance accomplishments as an exogenous variable.

**SC self-efficacy.** A 22-item scale measured trainees' confidence in their ability to complete specific SC tasks including scientific writing (10 items), presenting (4 items), and informal speaking (8 items; [13]). Ratings were obtained at four time points using a 5-point scale, ranging from 1 (*very insecure*) to 5 (*very confident*). In confirmatory factor analyses, nine items which did not load well were excluded from the scale. An item level confirmatory factor analysis at Time 1 demonstrated an acceptable fit to our current data ($\chi^2_{S-B}$ [62] = 96.542, $p <$ .01, CFI = .960, TLI = .950, RMSEA = .055; 90% CI [.032, .076], SRMR = .046). Cronbach's alpha provided evidence of acceptable internal consistency across all time points with good values for writing, .82-.86 (T1-T4), and presenting, .85-.89 (T1-T4), and acceptable values for informal speaking, .72-.76 (T1-T4). Subscale scores or parcels (i.e., averages of subscale items) were used in the analysis [40].

**SC outcome expectations.** A 6-item scale was used to assess trainee expectations of positive outcomes associated with performance in SC (writing, presenting, informal speaking; [13]), such as "allow me to obtain a highly desirable academic faculty position." Ratings were obtained at four time points using a 5-point scale, ranging from 1 (strongly disagree) to 5 (strongly agree). In preliminary analyses, one item was removed. The final item level confirmatory factor model with one residual covariance at Time 3 for the current study provided a good fit with the data ($\chi^2$S-B [4] = 4.752, p> .05, CFI = .993, TLI = .983, RMSEA = .035, 304 90% CI [.000, .132], SRMR = .025). Cronbach's alphas across all time points ranged from .73 to .81 (T1-T4).

**Science identity.** Identity as a scientist was measured at Time 2–4 with five items on a 5-point scale, ranging from 1 (*strongly disagree*) to 5 (*strongly agree*). Chemers' science identity scale was used [9], which includes items such as "I have a strong sense of belonging to the community of scientists." After removing one item which did not perform well, an item-level confirmatory factor analysis at Time 2 provided a good fit to the current data, $\chi^2_{S-B}$ (2) = 2.619, $p >$ .01, CFI = .996, TLI = .987, RMSEA = .045, 90% CI (.000, .172), SRMR = .019. Cronbach's alphas across all time points ranged from .86 to .87 (T2-T4).

**Mentoring practices in SC skill development.** At Time 1, a 12-item scale was used to assess trainees' perceived support from their mentors in mentoring behaviors specific to communication skills [31]. This scale measured mentoring practices in writing (5 items), presenting (4 items), and informal speaking (3 items) on a 5-point Likert scale, ranging from 1 (*never*) to 5 (*always*). Three items were excluded from the scale due to low loading on the factors. An item level confirmatory factor analysis demonstrated an adequate fit across most of the fit indices, $\chi^2_{\text{S-B}}$ (24) = 49.789, $p < .01$, CFI = .950, TLI = .925, RMSEA = .078, SRMR = .046. Cronbach's alpha was .74 for writing, .90 for presenting, and .64 for informal speaking. The three subscale scores were used for the analysis (writing, presenting, and informal speaking).

**Research career intention.** Three intention items assessed preferred role at Time 4. One item assessed a research role as a leader, "conduct and lead my own research and research team." One item assessed a support role, "support science and research, but not conduct research." The support item was negatively correlated with the first item, so it was reverse-scored to similarly reflect conducting research. A third item measuring participation in "conducting research, but not as the leader" was not related to the two intention items. Thus, the third item was dropped, although a two-item measure was not preferred. Items were measured on a 5-point scale from 1 (*strongly disagree*) to 5 (*strongly agree*). Cronbach's alpha was .56, but both construct reliability (.87) and construct validity (52%) were acceptable (coefficient H; [41]).

## Statistical analysis

We evaluated our structural model with latent variables over four time points using structural equation modeling (SEM) conducted with M*plus* Version 7.4 [42]. We used SEM because it is a modeling technique that allows us to capture simultaneous differential effects and predictive relationships of a variety of factors. Before fitting the hypothesized structural model, the measurement model was evaluated to ensure that all six latent variables were adequately represented by their items or subscales (parcels). Use of item parceling for a subscale in structural equation modeling yields as good as or slightly better results than does the use of item level as an indicator [40, 43]. The six latent factors were allowed to co-vary, where SC productivity was represented by 4 items, SC self-efficacy was represented by 3 subscales, SC mentoring practices by 3 subscales, science identity by 4 items, SC outcome expectations by 5 items, and career intention by 2 items.

Multiple indices were used to assess model fit, including the Satorra-Bentler chi-square ($\chi^2_{\text{S-B}}$), comparative fit index (CFI), Tucker-Lewis index (TLI), root-mean-square error of approximation (RMSEA), and standardized root-mean-square residual (SRMR). CFI and TFI values $\geq .95$, RMSEA values $\leq .06$, and SRMR values $\leq .08$ were used to indicate a good fit of the data [44].

## Results

### Preliminary analyses

**Missing data.** The attrition rate over four time points was 16%. We conducted an analysis to determine whether the data were missing completely at random, which would indicate that the sample was not biased (MCAR test; [40]); results suggested that this was the case, $\chi^2(300) = 265.647$, $p = .92$. Thus, we used maximum likelihood estimation with robust standard errors, which handles non-normal and missing data for analysis.

**Demographic information.** The participants reported their demographic variables, such as gender, first generation to attend college, academic rank, ethnicity, and discipline, as well as native language status. Gender was coded as "0" if the trainee was female, and "1" otherwise.

**Table 3. Correlations between indicators of SC model and trainees' demographic information.**

| | | Academic Rank | Sex | URM status | Generation | Primary Language | Discipline |
|---|---|---|---|---|---|---|---|
| | | 0 = Doctoral | 0 = Female | 0 = URM | 0 = 1st Gen | 0 = English | 0 = Basic Science |
| | | 1 = Postdoctoral | 1 = Male | 1 = WRM | 1 = Legacy | 1 = Other | 1 = Other |
| SC Productivity | Co-authored manuscripts | .53** | .06 | .08 | -.04 | .15* | .11 |
| | First-author manuscript | .58** | .14 | .08 | -.11 | .20** | .10 |
| | Oral presentation | .35** | .13 | .05 | .03 | .14 | .09 |
| | Poster | .40** | -.04 | -.05 | -.09 | -.02 | -.12 |
| SC Mentoring practices | Writing | .16* | .07 | -.11 | -.11 | .21** | .06 |
| | Presenting | -.02 | -.00 | -.04 | -.14 | .03 | -.13 |
| | Informal speaking | .04 | .05 | -.05 | -.03 | .17* | .01 |
| SC Self-efficacy | Writing | .20** | .05 | .10 | .03 | -.11 | .16* |
| | Presenting | .20** | .10 | .00 | -.09 | -.02 | -.06 |
| | Informal speaking | .02 | .17* | .07 | -.01 | -.04 | -.02 |
| SC Outcome expectations | | .08 | -.08 | -.17* | -.00 | .13 | -.02 |
| Science identity | | .08 | -.03 | .14 | -.03 | .10 | -.11 |
| Intend to be a Principal Investigator | | .31** | .12 | .07 | -.01 | .18* | .12 |
| Intend to support science and research, but not conduct research (reversed) | | .17* | .17* | -.03 | .05 | .03 | .10 |

\* and \*\* represent p < .05 and < .01, respectively. Positive coefficients mean that groups with a value = of "1" had a higher value in an indicator than did groups with a value of "0."

First generation was coded as "0" if the trainee was first generation, and "1" otherwise. The academic rank was coded with doctoral trainee as the reference group (0 = doctoral trainee; 1 = postdoctoral fellow). Ethnicity categories were categorized into underrepresented minority (URM; i.e., Hispanic, or Latino/Latina, American Indian/Native American, and African American) and well-represented (0 = URM; 1 = non-URM). The language status was coded with native speakers of English as the reference group (0 = English speaker; 1 = other language speaker). Discipline of a trainee's doctoral program was categorized into basic science and other (0 = basic science; 1 = other).

**Demographic differences.** We examined correlations among variables of interest and demographic variables (Table 3). In general, postdoctoral trainees had higher SC productivity and higher career intentions than did doctoral trainees, which was expected given their career stage. Compared with native speakers, non-native speakers of English reported higher writing productivity and higher intention to become an independent investigator. Male trainees reported higher self-efficacy in informal speaking and lower intention to support science (not conduct research). Trainees of underrepresented groups reported higher outcome expectations than did trainees of well-represented groups. There were no significant differences in any variables of interest between basic science and others or between first-generation and continuing generation trainees.

Multiple regression analyses, however, indicated that the three demographic variables (i.e., gender, rank, and language status) that were correlated with at least one of the intention indicators did not significantly predict the two career intention indicators, when controlling for SC productivity, SC outcome expectations, and science identity.

## Measurement model

The test of the six-factor measurement model, including latent variables of SC productivity, SC self-efficacy, SC mentoring practices, science identity, SC outcome expectations, and

research career intention demonstrated an adequate fit to the data, $\chi^2_{S-B}$ (173) = 255.417, $p <$ .01; CFI = .954; TLI = .944; RMSEA = .041, 90% CI (.024, .055); SRMR = .063. In addition, most indicators had relatively high standardized factor loadings (e.g., > .60), showing that the indicators of the six latent variables adequately measured each construct.

## Structural model

Analysis of the hypothesized structural model provided an adequate fit to the data, $\chi^2_{S-B}$ (178) = 226.512, $p <$ .01; CFI = .957; TLI = .949; RMSEA = .039, 90% CI (.021, .053); SRMR = .065. Path coefficients for the structural model are shown in Fig 1. As hypothesized, the four new paths were statistically significant; that is, SC mentoring practices significantly predicted SC self-efficacy (path b, $B$ = .479, $p <$ .001) and SC outcome expectations (path f, $B$ = .241, $p$ = *.025*). In addition, SC self-efficacy significantly predicted science identity (path d, $B$ = .394, $p <$ .001) and science identity significantly predicted SC outcome expectations (path h, $B$ = .427, $p <$ .001).

Consistent with our previous cross-sectional study, trainees' SC productivity significantly predicted SC self-efficacy (path a, $B$ = .421, $p <$ .001) and career intentions (path c, $B$ = .378, $p$ = .001), and SC outcome expectations significantly predicted career intention (path i, $B$ = .338, $p$ = .004). Contrary to our hypotheses, SC self-efficacy did not significantly predict SC outcome expectations (path e, $B$ = -.212, $p$ = .093). Science identity did not directly predict career intention (path g, $B$ = .131, $p$ = .259).

We also examined the indirect effects of the five variables on career intention by running 5000 bias-corrected bootstrap samples in M*plus*. As hypothesized, three indirect paths yielding significant pathways included the path from SC mentoring practices to science identity via SC self-efficacy (b→d; indirect effect = .248, 95% CI [.119, .433]), the path from SC productivity to science identity via SC self-efficacy (a→d; indirect effect = .112, 95% CI [.052, .210]), as well as the path from science identity to intention via SC outcome expectations (h→i; indirect effect = .212, 95% CI [.074, .480]). In addition, SC productivity indirectly predicted career intention through SC self-efficacy, science identity, and SC outcome expectations (a→d→ h→I; indirect effect = .024, 95% CI [.007, .072]). SC self-efficacy indirectly predicted career intentions through science identity and SC outcome expectations (d→h→I; indirect effect = .161, 95% CI [.047, .459]). Mentoring practices indirectly predicted career intention through SC outcome expectations or through SC self-efficacy, identity, and SC outcome expectations (f→i or f→b→d→h→i; indirect effect = .144, 95% CI [.025, .397]). In addition, SC self-efficacy indirectly predicted SC outcome expectations through science identity (d→h; indirect effect = .229, 95% CI [.098, .464]).

The SC model accounted for relatively large amounts of the variance in SC self-efficacy (41%) and career intention (32%) and more modest amounts of the variance in science identity (15%) and SC outcome expectations (20%).

## Discussion

As the first longitudinal test of the impact of scientific communication on career intention, the SC model explained the notable degree to which SC skills contribute to intention to pursue a research career—32%—at the doctoral and postdoctoral level. We conclude that engaging in SC and receiving active mentoring in SC skills play a significant role in intention to persist in academic research careers at the PhD and postdoctoral level. The three modes of scientific writing, presenting, and spontaneous speaking (conversation) each contributed to these effects. While quality of writing and speaking is undoubtedly important, the effects reported here stemmed from frequency of engagement. Similarly, effects of mentoring practices

measured were based on engagement in mentoring, not on evaluations of quality or on special-ized mentoring techniques. Taken together, these effects suggest that attending to and encour-aging trainees to engage more in all forms of communication can help to strengthen their science identity and their intentions to persist in research.

## Strengths and limitations

In addition to providing evidence for the developmental role of language use in fostering sci-ence identity during research training and, ultimately, in influencing career intention, the strengths of this study include the longitudinal design, infrequent among SCCT studies. Our study used data from four time points, enabling analysis of how the mechanisms of interest developed over time and capturing predictive relationships among the variables. In addition, this study addressed career intentions at the postgraduate level, adding valuable perspectives to the growing scholarship on research career intentions and persistence beyond the undergradu-ate level. Our focus has been on language use at the doctoral and postdoctoral levels, when the imminent prospect of a research career begins to raise concerns about producing peer-reviewed publications and oral presentations, but a similar influence of SC on career intention at earlier stages of education is plausible. Finally, this study makes an important contribution to sociolinguistic scholarship by examining linguistic principles from an evidence-based and theory-driven social-cognitive perspective.

Some limitations of this study should be considered when interpreting the findings. First, selection bias may have occurred since the survey respondents could have had specific interest in the study content. Second, there are measurement and statistical limitations. In previous studies, we developed our social cognitive measures specific to scientific communication in samples of doctoral and postdoctoral trainees (e.g., self-efficacy, outcome expectations, men-toring practices [13, 31]). In the current study, psychometric properties for some of these mea-sures were less than optimal. Further item development and measure evaluation is needed with larger samples to better capture the model constructs on which we have focused as well as their relationships. In particular, the speaking factor of mentoring practices and the measure of career intention warrant further item development. Future research will examine group dif-ferences by race/ethnicity and other relevant subgroups. Given adequate subgroup sample size, this should provide a more detailed understanding of scientific communication dynamics in various populations. This is especially relevant, for example, to participants who were raised speaking stigmatized regional or social dialects of English. Finally, this study did not examine the potential role of reciprocal effects, which have been shown in multiple structural equation modeling studies [45] to emerge over extended periods of time. Reciprocal effects between lan-guage use and identity are likely, as discussed in the next section.

The results of this study have implications not only for efforts to broaden and strengthen participation in research careers, but also for social cognitive career theory and for the socio-linguistic scholarship concerning identity. We discuss each of these below.

## Relationship of language use and identity development in research training to entering the community of practice

From a sociolinguistic perspective, our study provides empirical evidence for a psychological link between oral and written communication and individual identity and is, to our knowl-edge, the first to show that language use (in this case, SC productivity) may be a precursor of identity. Sociolinguistic research on identity has traditionally used approaches based on *post-hoc* ethnographic analysis of conversations or other speech artifacts that are considered to reflect identity [14, 15], or of observation of listener reactions to the speech of others with the

aim of investigating how listeners attribute ethnic or social identity to a speaker [46]. Some social-psychological studies have provided empirical evidence for associations between identity and dynamics of second language acquisition and usage [47] or for the use of language as a tool to assert ethnic identity [47, 48]. To our knowledge, the findings reported here are among the first to examine social-psychological processes underlying *individual* identity development and its sources in language use.

In addition to the individual psychological processes of identity development through language production, the relationality principle in socio-cultural linguistics holds that identity acquires meaning through social interaction. In other words, identity is meaningless in isolation from other people. As others communicate to the individual, they are both modeling the group's particular linguistic style and characteristics as well as affirming or challenging the individual's membership in the group, whether implicitly or explicitly. In the scientific research environment, the group in question can be understood as the research "community of practice," as described by Lave and Wenger [49]. (A community of practice is defined as a "group of people who share a concern. . .for something they do and learn how to do it better as they interact regularly" [50].) The goal of training is to be accepted into this community and recognized as a legitimate practitioner. As research trainees develop their communication skills, they do so by imitating the language of senior researchers and peers in the environment and then receiving feedback that authenticates their identities as members of the research community (or fails to do so; [21]). A quote from a doctoral-level focus group participant from a previous study provides a concrete example: ". . . I feel that people are "smarter" than I am or have access to a broader range of vocabulary than I have, particularly around scientific conversations . . . I wanted to quit . . . so [I'm] trying to learn to speak the way they do, to write the way they do, to act the way they do—really to assimilate myself in this particular culture" [51].

As this student's words imply, when speakers demonstrate mastery of the language variety of a particular group, they are more likely to feel accepted as members of the group and judged more favorably. With respect to STEM research training, all trainees must acquire the professional-level language of their discipline in order to succeed. Our findings, demonstrating the role of language use and mentor support in science identity development and career intention, serve as an initial test of the relationality principle. Future studies may find evidence for a fully reciprocal relationship between language use and identity—a feedback loop—by investigating the influence of science identity on trainees' continued engagement in scientific communication.

Taken together, the empirical evidence presented here for the role of language use as a source of identity and the role of SC mentoring as an indirect source of identity are key quantitative contributions to sociolinguistic scholarship on identity and language use as well as to the study of the individual's socialization into communities of practice.

### Relevance to broadening participation in research careers

Additional questions are raised regarding intersectional implications for trainees who were raised speaking regional, ethnic, or social dialects of English, such as Spanish-influenced English, African-American Vernacular English, Appalachian or southern dialects, urban or rural working-class dialects, etc. Such varieties are frequently perceived by society as inferior to standard English, and in more extreme cases, as negative indicators of character or intellect. A study of attitudes about dialect diversity conducted among undergraduate students at the University of North Carolina reported that 61% of respondents agreed that "the way they speak is an important part of their identity." Qualitative data indicated awareness that non-

standard dialects are often perceived more negatively *or* more positively than standard English [52]. While the vast majority of graduate students and postdocs in STEM use standard English in the research environment, preliminary evidence from this study (reported in [31]) suggests a possible relationship between having been *raised* speaking a non-standard dialect and lower perceptions of belonging in the research environment. Some demographic subgroups in this study who endorsed having been raised speaking a non-standard dialect reported lower sense of belonging in the research environment, which Gibbs [6] found to be associated with declining levels of interest in pursuing a research career among PhD and postdoctoral URM women. The paired mentors of trainees speaking non-standard dialects viewed these mentees as presenting higher barriers to mentoring SC in some categories, despite being unaware (to our knowledge) of the mentee's linguistic background. We continue to collect additional data needed to investigate these possible relationships, as well as the nature of the relationship between identity, sense of belonging, and entry into the research community of practice. If more definitive conclusions are reached, tailored interventions should be considered.

## Implications for social cognitive career theory

The results of this study extend current scholarship on social cognitive career theory. Motivated by the links between language use and identity discussed above, we extended our cross-sectional model to a longitudinal design, and included science identity and SC mentor practices. Earlier SCCT models have not included science identity, instead linking self-efficacy directly to outcome expectations [38]. The results we report here, however, support only a significant, indirect path from self-efficacy through science identity to SC outcome expectations. In contrast, recent work by Byars-Winston and Rogers [36] found a direct relationship between research self-efficacy and outcome expectations among undergraduates, as well as indirect relationships through science identity. Our results suggest that there may be different relationships between self-efficacy and outcome expectations for doctoral and post-doctoral trainees in the specific domain of scientific communications. Consistent with Byars-Winston and Rogers [36], we found that sources of learning that form self-efficacy beliefs [34] were associated with the SCCT model's social-cognitive constructs in accounting for research career intention. Notably, the three sources included in our model (i.e., performance accomplishments by trainees, vicarious experience and social persuasion from mentors) explained the variance in self-efficacy in scientific communication well, emphasizing the importance of these experiences to SC self-efficacy development.

Including science identity is a new feature in the SCCT model, especially for doctoral and postdoctoral trainees, so these findings should be further investigated by evaluating variable relationships across ethnically diverse groups over time. We hypothesized that science identity predicts career intentions directly or indirectly. Our findings support only an indirect relationship, since SC outcome expectations fully mediated the effect of science identity on career intentions. This suggests that science identity may help doctoral and postdoctoral trainees elaborate career outcome expectations related to engaging in SC, which then supports sustained intention to pursue a research career.

## Implications for practice in STEM research training and mentoring

Our study of the influence of SC skills on intention to persist in a research career has important implications for research training. If science identity is a significant predictor of career intentions and if language use is a precursor to identity, the encouragement and reinforcement of trainee engagement in SC may be a key strategy for increasing career persistence through the postgraduate and postdoctoral levels. Moreover, raising awareness of the potential impact

of language variety may eventually prove to be a new means of addressing diversity and inclusion in the research environment.

Mentors can benefit from becoming aware of the critical role of communication skills in research training and learning a variety of strategies to stimulate it. Research mentor training in general has been shown to be feasible and effective [37, 53, 54]; mentor training for fostering the development of SC skills, as suggested by the role of SC mentoring practices in the development of self-efficacy, is feasible as well. Strategies that mentors can acquire to increase SC self-efficacy, science identity, and outcome expectations in their trainees include acknowledging and communicating SC's impact, providing structured expectations for engaging in SC, stimulating vigorous engagement in scientific writing, speaking, and presenting, and providing skillful and appropriate feedback. The effectiveness of applying such techniques in mentors' professional development and in their mentees' self-efficacy, identity, and career intention is currently being tested in SC mentor skills workshops at four sites nationally (R25 GM125640, C. Cameron & S. Chang, MPIs). Such training may impact not only trainees' career intention, but ultimately their success as well, through increased ability to disseminate research findings.

## Final thoughts

The causes of attrition in STEM research careers are multiple and complex. Our study focused on a novel sociocultural and linguistic influence on intention to persist in research careers through engagement in SC skills as they relate to inclusion in the research community of practice. Recognition of the significant role of SC opens up new perspectives on the potential of SC as an area for further research and intervention. Such research would illuminate the mechanisms by which specific aspects of SC contribute to intention and test the relative value of specific SC mentoring strategies. If future findings suggest a differential impact of these variables on members of underrepresented groups, new approaches to increasing their retention in research careers may be possible. Beyond the relevance of SC for STEM careers, appreciation of the role of communication skills in identity development in general suggests the possibility for numerous lines of inquiry and application in the social sciences, pedagogy and education theory, diversity and inclusion efforts, and professional development fields as well.

## Supporting information

**S1 Data.**
(XLS)

## Acknowledgments

The authors are grateful to Paul Hernandez for valuable commentary on the analysis; to Tamara Locke and the MD Anderson Department of Scientific Publications for editing assistance; and to Erin K. Dahlstrom and Yen Nhi Pham for project assistance.

## Author Contributions

**Conceptualization:** Carrie Cameron, Shine Chang.

**Data curation:** Hwa Young Lee.

**Formal analysis:** Hwa Young Lee, Cheryl B. Anderson.

**Funding acquisition:** Carrie Cameron, Shine Chang.

**Investigation:** Hwa Young Lee, Cheryl B. Anderson.

**Methodology:** Carrie Cameron, Hwa Young Lee, Cheryl B. Anderson, Shine Chang.

**Project administration:** Hwa Young Lee, Jordan Trachtenberg.

**Supervision:** Carrie Cameron, Cheryl B. Anderson.

**Validation:** Cheryl B. Anderson.

**Writing – original draft:** Carrie Cameron, Hwa Young Lee, Cheryl B. Anderson.

**Writing – review & editing:** Hwa Young Lee, Cheryl B. Anderson, Shine Chang.

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
