## [Decision Letter · Decision Letter 0]

11 Nov 2019

PONE-D-19-15861

The role of scientific communication in predicting science identity and research career intention

PLOS ONE

Dear Dr. Cameron,

Thank you for submitting your manuscript to PLOS ONE. After careful consideration, we feel that it has merit but does not fully meet PLOS ONE’s publication criteria as it currently stands. Therefore, we invite you to submit a revised version of the manuscript that addresses the points raised during the review process.

Both reviewers provided several constructive comments to improve the manuscript. I would encourage the authors to address each of these in a revised version.

We would appreciate receiving your revised manuscript by Dec 26 2019 11:59PM. To enhance the reproducibility of your results, we recommend that if applicable you deposit your laboratory protocols in protocols.io, where a protocol can be assigned its own identifier (DOI) such that it can be cited independently in the future. For instructions see: http://journals.plos.org/plosone/s/submission-guidelines#loc-laboratory-protocols

We look forward to receiving your revised manuscript.

Kind regards,

Cassidy Rose Sugimoto, Ph.D.

Academic Editor

PLOS ONE

Journal Requirements:

2.  Please provide additional details regarding participant consent. In the ethics statement in the Methods and online submission information, please ensure that you have specified whether consent was informed. If the need for informed consent was waived by the ethics committee, please include this information.

Reviewers' comments:

Reviewer's Responses to Questions

**Comments to the Author**

1. Is the manuscript technically sound, and do the data support the conclusions?

Reviewer #1: Yes

Reviewer #2: Partly

2. Has the statistical analysis been performed appropriately and rigorously? 

Reviewer #1: I Don't Know

Reviewer #2: Yes

3. Have the authors made all data underlying the findings in their manuscript fully available?

Reviewer #1: Yes

Reviewer #2: Yes

4. Is the manuscript presented in an intelligible fashion and written in standard English?

Reviewer #1: Yes

Reviewer #2: Yes

5. Review Comments to the Author

Reviewer #1: Summary and overall impression: The article makes a nice contribution to the understanding of career thinking among biomedical trainees by calling attention to scientific communication and its relationship to science identity. This is novel and useful research. It is well-written with narrative that corresponds nicely with the charts and figures. The integration of scientific communication constructs within SCCT will be helpful to anyone concerned with supporting and developing biomedical PhD students

Areas for improvement: I found the article well-written and clear, so my comments fall into the “minor issue” category. I hope that attention to these points will improve an already strong paper:

a. Because career intention is an important focus of this paper, I suggest some attention to clarifying and staying consistent with terminology. I see “research career” in the title, “academic faculty career” in the abstract, “research career intention” (p. 16), and “academic career intention” used in Figure 1. Do these all refer to a faculty career that is research-intensive, i.e., a PI? As I attempted to seek clarification, I reread the text section on p. 16 and then went to the table 3 – on table 3 I saw “intend to be a PI” and “intend to work in a higher ed academic setting” in table 3, which were not consistent with the text description on p. 16.

b. With science identity as a central focus, I wanted to know the items used to measure this construct with a bit more detail than provided. I feel the claims about science identity rest on this measurement of science identity thus necessitating the addition of details of what aspects of science identity were measured.

c. In Table 2, first-gen college may need a definition (neither parent attained a bachelor’s degree?) as sometimes first-gen is defined differently. As well, I was curious what was used for the “economic standing” characteristic, although that variable was not mentioned in the text so it may not be necessary to include on the table.

d. When I saw the focus group quotation on p. 23, I did not recall seeing a mention of focus groups in the methodology section. Was this part of the current study or drawn from earlier research?

e. “Communities of Practice” may need citation (Lave & Wenger’s work, 1991) somewhere in the introductory/theoretical framing sections because this concept appears in the discussion section several times (p. 24, 30). Though an alternative is to reword without using the specific phrase, community of practice.

f. “given that science identity is a significant predictor of career intentions” (p. 28/29) – may be too strong a statement as even some studies cited previously in the paper (Estrada, et al.) on p. 9 “did not predict career outcomes.”

Other: Given that trainees often bristle at critical feedback and may feel singled out, I think another audience for this work is the trainees themselves. The idea of scientific communication as a distinct language variety that can improve through practice and frequency is powerful: it normalizes language development in a particular field as a process that everyone goes through to be recognized in that field, both by oneself and by others.

I am not an expert in statistics (which is why I answered "do not know" for the statistical analysis question in this review), but given that the statistics are accurate, I found it easy to follow the findings because of the good textual descriptions that corresponded with Table 3 and Figure 1.

Reviewer #2: Strengths

• Much of the research and interventions that target STEM careers have focused on undergraduates with the goal of increasing the number of students, especially women and underrepresented minorities who pursue doctoral degrees. The erroneous assumption made here is that once individuals reach the doctoral or postdoctoral phase, pursuing a research career is the most likely or the most desired career pursuit. However, the data do not support this assumption as the proverbial “leaks” become more pronounced as individuals advance through the academic pipeline. Thus, the current study is an important contribution as it targets doctoral and postdoctoral trainees who are significantly underrepresented in STEM/biomedical research intervention studies.

• Additional areas of strength are the longitudinal design, sample diversity in terms of gender, race/ethnicity, and first generation status, and the inclusion of science specific constructs rather than focusing only on social identity constructs.

• Additional strength is the inclusion of scientific communication (and its operationalized components) within a broader model of intentions to pursue research career.

• The conceptual model presented in theoretically grounded and includes several well established and supported science intention constructs (e.g., science self efficacy, science identity) as well as social behavioral theories that show desired outcomes are more likely when expectancies and behavioral skills approximate intentions (e.g., SC skills, SC products, SC outcomes).

Limitations

• The study includes both doctoral trainees and postdoctoral students with the doctoral students outnumber postdocs 3 to 1. As one would expect, these are two very different groups especially in light of the study constructs—intentions to pursue research career and research products (presentations and publications). The authors do not provide data on the developmental stage of the doctoral students (i.e., average year in their doctoral program). Doctoral students in their first or second year will likely have fewer first author publications and conference presentations than more advanced students. It is unclear if the doctoral student sample represents a broad distribution in terms of years in program or are skewed either early or more advanced graduate students.

• The authors stated they recruited biomedical and behavioral sciences programs. Additional detail should be provided for how disciplines were included for recruitment especially in light of 25% of the sample falling into the “other” category. For instance, were clinical psychology programs included which provide training in both research and clinical practice versus behavioral science programs that focus only on research training?

• The authors posit a clear unidirectional framework that language skills precede and lead to identity—based primarily on social/cultural and cognitive theories of language development. However, there are some limitations to applying this framework to the current model of scientific communication and science identity. A science identity is one that is acquired and completely volitional which is not the case for many social identities (race, gender, cultural) where language precedes identity formation. Is it not possible that in the context of scientific career intentions, the relationship between scientific communication and science identity is bidirectional instead of unidirectional? Is it not possible for a science identity to develop before one has acquired scientific communication skills and that science identity facilitates (or motivates) one to acquire SC skills? Moreover, the current sample includes doctoral students and postdocs in biomedical research disciplines, which one could assume already have some degree of science identity prior to entering their doctoral programs, yet science identity is not measured at Time 1.

• This is a longitudinal dataset but it is unclear why some constructs were measured at multiple time points and others at single time points. For instance, SC productivity was measured only at Time 1 and SC outcome expectations at Time 3. Why would we not expect SC productivity to increase over time as participants advance through their training programs and that productivity may directly predict more proximal SC outcomes (Time 1 experiences predict Time 2 outcomes, etc). And what about length of time in program as a covariate for SC Productivity---productivity more likely for advanced doctoral students.

• The authors point to the dyadic relationship (mentor-trainee) as a novel contribution to the current study but only present the trainee data rather than dyadic analyses.

• There are a couple of discrepancies between the variables in Table 3 and the description in the methods section. In the text women are coded as 1 but as 0 in the table. Similarly, in the text first gen is coded as 1 but as 0 in the table.

• The authors also make a couple of conclusions in the discussion that did not appear to be reflected in their analyses. For instance in the following sentence, “Scientific writing, presenting, and spontaneous speaking (conversation) all contributed to these effects, and effects were not dependent on the perceived “quality” of the skills or success in publication. Rather, they stemmed from frequency of engagement.” It is not clear to me how the authors can discount “quality” when they did not report measuring quality but only measured frequency (and limited frequency with some measures due to item/scale limitations).

• The authors did address some of the scale limits in their discussion; however, this could be expanded a bit more. Each of the SC scales had items that did not load on derived factors and were subsequently omitted from analyses. The most problematic of which were SC self-efficacy with 40% of the items not loading and SC outcome expectations with 50% of the items not loading. Was there limited prior validity work using these scales or something about the current sample to explain the scale differences?

6. PLOS authors have the option to publish the peer review history of their article (what does this mean?). If published, this will include your full peer review and any attached files.

Reviewer #1: Yes: Robin Remich

Reviewer #2: No

---

## [Author Response · Author response to Decision Letter 0]

4 Dec 2019

Responses to each point raised by the academic editor and reviewers are included in the document Response to Reviewers.

---

## [Decision Letter · Decision Letter 1]

10 Jan 2020

The role of scientific communication in predicting science identity and research career intention

PONE-D-19-15861R1

Dear Dr. Cameron,

We are pleased to inform you that your manuscript has been judged scientifically suitable for publication and will be formally accepted for publication once it complies with all outstanding technical requirements.

With kind regards,

Cassidy Rose Sugimoto, Ph.D.

Academic Editor

PLOS ONE

Additional Editor Comments (optional):

Reviewers' comments:

Reviewer's Responses to Questions

**Comments to the Author**

1. If the authors have adequately addressed your comments raised in a previous round of review and you feel that this manuscript is now acceptable for publication, you may indicate that here to bypass the “Comments to the Author” section, enter your conflict of interest statement in the “Confidential to Editor” section, and submit your "Accept" recommendation.

Reviewer #1: All comments have been addressed

Reviewer #2: All comments have been addressed

2. Is the manuscript technically sound, and do the data support the conclusions?

Reviewer #1: (No Response)

Reviewer #2: Yes

3. Has the statistical analysis been performed appropriately and rigorously? 

Reviewer #1: (No Response)

Reviewer #2: Yes

4. Have the authors made all data underlying the findings in their manuscript fully available?

Reviewer #1: (No Response)

Reviewer #2: Yes

5. Is the manuscript presented in an intelligible fashion and written in standard English?

Reviewer #1: (No Response)

Reviewer #2: Yes

6. Review Comments to the Author

Reviewer #1: (No Response)

Reviewer #2: The authors have adequately addressed the concerns raised by the previous reviews. There is one minor issue that was missed in the previous reviews. On page 9 the authors state, "similarly, we included science identity as a potential mediator between SC self-efficacy and career intention in our longitudinal model." However, this prediction was not included among the hypotheses listed on page 10 although the pathway was tested in the SEM analyses.

7. PLOS authors have the option to publish the peer review history of their article (what does this mean?). If published, this will include your full peer review and any attached files.

Reviewer #1: Yes: Robin Remich

Reviewer #2: No

---

## [Editor Report · Acceptance letter]

24 Jan 2020

PONE-D-19-15861R1 

The role of scientific communication in predicting science identity and research career intention 

Dear Dr. Cameron:

I am pleased to inform you that your manuscript has been deemed suitable for publication in PLOS ONE. Congratulations! Your manuscript is now with our production department. 

With kind regards,

on behalf of

Dr. Cassidy Rose Sugimoto 

Academic Editor

PLOS ONE